# Circular Intronic RNA circTTN Inhibits Host Gene Transcription and Myogenesis by Recruiting PURB Proteins to form Heterotypic Complexes

**DOI:** 10.3390/ijms24129859

**Published:** 2023-06-07

**Authors:** Nini Ai, Zonggang Yu, Xueli Xu, Sui Liufu, Kaiming Wang, Shengqiang Huang, Xintong Li, Xiaolin Liu, Bohe Chen, Haiming Ma, Yulong Yin

**Affiliations:** 1College of Animal Science and Technology, Hunan Agricultural University, Changsha 410128, China; 15291270712@163.com (N.A.); study236@163.com (Z.Y.); xxl1632501152@163.com (X.X.); liufusui0816@163.com (S.L.); m15116529648@163.com (K.W.); hsq07@126.com (S.H.); p13548614212@163.com (X.L.); lxl810711@163.com (X.L.); chenhe0213914@163.com (B.C.); 2Guangdong Laboratory for Lingnan Modern Agriculture, Guangzhou 510642, China; 3Institute of Subtropical Agriculture, Chinese Academy of Sciences, Changsha 410125, China

**Keywords:** circTTN, C2C12, myogenesis, proliferation, PURB, *TTN*

## Abstract

Muscle cell growth plays an important role in skeletal muscle development. Circular RNAs (circRNAs) have been proven to be involved in the regulation of skeletal muscle growth and development. In this study, we explored the effect of circTTN on myoblast growth and its possible molecular mechanism. Using C2C12 cells as a functional model, the authenticity of circTTN was confirmed by RNase R digestion and Sanger sequencing. Previous functional studies have showed that the overexpression of circTTN inhibits myoblast proliferation and differentiation. Mechanistically, circTTN recruits the PURB protein on the *Titin* (*TTN*) promoter to inhibit the expression of the *TTN* gene. Moreover, PURB inhibits myoblast proliferation and differentiation, which is consistent with circTTN function. In summary, our results indicate that circTTN inhibits the transcription and myogenesis of the host gene *TTN* by recruiting PURB proteins to form heterotypic complexes. This work may act as a reference for further research on the role of circRNA in skeletal muscle growth and development.

## 1. Introduction

Skeletal muscle is an important component of the mammalian body, which plays an important role in energy transformation and posture maintenance [1]. The skeletal muscle development process includes three stages: (1) mesenchymal stem cells from mesoderm proliferate and differentiate into myoblasts, (2) myoblasts further differentiate to form primary myotubes and secondary myotubes, and (3) myotubes then fuse into muscle fibers [2]. This process is also known as myogenesis. Muscle fibers are the basic contractile units of skeletal muscle. They are surrounded by connective tissue layers and are formed into bundles to form skeletal muscle [3]. Skeletal muscle development is subject to myogenic regulatory factors (*MRFs*), paired box family (*PAX*), SIX family, and myocyte enhancer factor 2 (*MEF2*) [4,5]. In recent years, with the upsurge of research on non-coding RNAs (ncRNAs), the mechanism of circular RNAs (circRNAs) in skeletal muscle growth and development is also attracting researchers’ interest.

circRNAs are a class of non-coding RNA molecules that do not have a 5′ cap and 3′ poly(A) tail. Dicer [6] first discovered the potato spindle tuber “virus” in 1971 and called it viroid. In 1993, circular transcription of the *Sry* gene was found in mice, but the circRNAs discovered were thought to be low-abundance RNA molecules formed by the missplicing of exon transcripts [7]. With the rapid development of high-throughput sequencing, more circRNAs have been discovered by researchers [8]. Currently, circRNAs have been shown to be associated with myogenesis, the transformation of muscle fiber types, and muscle diseases [9,10,11]. For example, circZfp609 can act as a molecular sponge to adsorb miR-194-5p and isolate its inhibition of *BCLAF1*, thus inhibiting myogenic differentiation [12]. In addition, circRNAs can regulate the variable splicing of transcripts and parental gene transcription and translation by binding to RBPs [13,14,15]. However, the previous research on this mechanism has mainly been focused on the aspect of cancer treatment and less on the aspect of skeletal muscle development, which requires further exploration.

We previously found a circular intronic RNA circTTN produced by the *TTN* (*Titin*) gene through high-throughput sequencing [16]. circTTN was significantly differentially expressed in the skeletal muscle of Ningxiang pig at 30 d, 90 d, 150 d, and 210 d after birth, suggesting that circTTN may be involved in the regulation of skeletal muscle myogenesis. In vitro, functional validation models (C2C12 cells) showed that circTTN is a negative regulator of myogenesis and that circTTN recruits PURB protein to inhibit the transcription of the host gene *TTN*. Our study revealed the molecular regulatory mechanism of circTTN and provided a reference for further research on skeletal muscle growth and development.

## 2. Results

### 2.1. Characterization Analysis of circTTN

circTTN is derived from intron15 of the *TTN* gene; its length is 339 bases, and it is located in chromosome 15 (Chr15_85761324_85761662) (Figure 1A). First, the divergent primer and convergent primer were designed to perform qRT-PCR experiments using the cDNA and gDNA of muscle tissue as templates, and DNA gel electrophoresis was performed on the products. The convergent primer and divergent primer could amplify bands using cDNA as a template. Only the convergent primer could amplify a band when gDNA was used as the template (Figure 1B). The products of the divergent primer were analyzed using Sanger sequencing (Figure 1A). After treatment with RNase R, *TTN* and *β-actin* genes had no bands after electrophoresis, while circTTN had bands (Figure 1C). The above experimental results can confirm the authenticity and stability of circTTN. 

Then, the expression of circTTN in different tissues of Ningxiang pig at four periods after birth were detected using qRT-PCR assay. The results showed that the expression of circTTN in the heart and muscle of Ningxiang pig at four stages was the highest (Figure 1D). In addition, circTTN was differentially expressed during the proliferation and differentiation of C2C12 cells (Figure 1E,F). 

### 2.2. circTTN Inhibits Proliferation and Differentiation of C2C12 Cells and Host Gene TTN

To explore the function of circTTN, the circTTN overexpression vector was transfected into C2C12 cells, and the transfection efficiency of circTTN was more than 3000 times higher (*p* < 0.01) (Figure 2A), which can be used for subsequent experiments. First, qRT-PCR and Western blot were used to detect the expression of proliferation biomarker genes. The overexpression of circTTN significantly reduced the mRNA levels of *CCND* and *CDK4* (*p* < 0.05); the mRNA expression level of *PCNA* was markedly decreased (*p* < 0.01) (Figure 2B). The protein expression levels of proliferating genes CCND and PCNA in the overexpression group were markedly lower than those in the control group (*p* < 0.01); the CDK4 protein expression level was significantly decreased (*p* < 0.05) (Figure 2C,D). The results of the CCK8 experiments displayed that there were no significant difference in the proliferative activity between the overexpressed group and the control group (*p* > 0.05) (Figure 2E); the EdU results showed that the number of positive proliferating cells in the overexpressed group was markedly decreased (*p* < 0.01) (Figure 2F,G). In addition, flow cytometry was used to detect circTTN expression during cell cycle process, and the number of cells in the overexpressed group was significantly reduced in the S phase (*p* < 0.01) (Figure 2H,I). In summary, circTTN inhibited the proliferation of C2C12 cells. 

We further explored the effect of circTTN on the differentiation of C2C12 cells; the overexpressed circTTN vector was transfected into cells for induction differentiation culture, and then, qRT-PCR and a Western blot assay were used to detect the expression of differentiation biomarker genes. The overexpression of circTTN significantly inhibited the expression of differentiation biomarker genes (*MyoD*, *MyHC*, and *MyoG*) at the mRNA level (Figure 3A) (*p* < 0.01). At the protein level, the expression of *MyoD* and *MyoG* were markedly repressed (*p* < 0.01), significantly inhibiting the expression of *MyHC* (*p* < 0.05) (Figure 3B,C). In addition, an immunofluorescence assay was used to observe the changes in phenotypic myotube differentiation, and the number in differentiated myotubes in the circTTN overexpression group was significantly reduced compared with the control group (*p* < 0.05) (Figure 3D). These results indicate that circTTN inhibited the differentiation of C2C12 cells.

*TTN* is a coding gene that plays an important role in skeletal muscle. We intended to discover if circTTN acts on host gene *TTN*. The qRT-PCR results implied that circTTN overexpression significantly downregulated *TTN* gene expression during the proliferation period (Figure 4A) (*p* < 0.01). At the protein level, the overexpression of circTTN was markedly lower during proliferation compared with the control group (Figure 4B,C) (*p* < 0.01). Similarly, the effect of circTTN overexpression on the *TTN* gene during differentiation was consistent with that during proliferation, and the mRNA and protein levels of circTTN were significantly decreased compared with those of the control group (*p* < 0.01) (Figure 4D–F).

### 2.3. circTTN Interacts with PURB Protein

First, we determined the subcellular localization of circTTN by nucleoplasmic isolation and RNA fluorescence in situ hybridization experiments. *GAPDH* and *U6* were used as reference genes in the cytoplasm and nucleus, respectively. The results showed that circTTN and *TTN* mRNA existed in the nucleus and cytoplasm, but most of them were located in the nucleus (Figure 5A). The results of the FISH experiment were consistent with the results of nucleoplasmic separation. (Figure 5B). These results suggest that circTTN may be regulated in the nucleus. circRNA can interact with RNA-binding proteins to influence the expression of parent genes. We then examined circTTN-bound proteins using an RNA pull-down assay. The circTTN group showed obvious bands of difference compared with the control NC group (Figure 5C). Subsequently, a total of 356 proteins were identified using mass spectrometry (MS) analysis, among which 218 different proteins were found in the circTTN group (RPD group) (Figure 5D, Table 1). Gene ontology (GO) functional enrichment analysis and Kyoto Encyclopedia of Genes and Genomes (KEGG) analysis were conducted, respectively. GO functional enrichment showed that these proteins were mainly enriched in cellular processes, protein complexes, regulatory translation activities, biological regulation, developmental processes, and binding. KEGG suggested that some proteins were enriched in transcription, translation, and metabolism (Figure 5F). These results implied that circTTN may bind to these proteins and play a certain function. Next, the expression of Pur-beta (PURB) protein in the RNA pull-down eluent was detected using a Western blot assay (Figure 5G), and it was found that the expression of PURB protein could be only detected in the circTTN group’s eluent. In addition, we reversely demonstrated an interaction between PURB protein and circTTN through an RIP experiment (Figure 5H). The results showed that the PURB antibody could be significantly enriched to circTTN (*p* < 0.01), further supporting the idea that circTTN interacts with PURB.

### 2.4. PURB Knockdown Promotes Proliferation and Differentiation of C2C12 Cells and Increase TTN Expression

Previously, we confirmed the interaction between PURB protein and circTTN. Next, we detected the expression of PURB in C2C12 cells using a Western blot assay (Figure 6A), and it was found that PURB was highly expressed in C2C12 cells. To further explore the effect of PURB on C2C12 cells, we designed 3 siRNA sequences, as shown in Figure 6B; qRT-PCR was used to select the one with the best knockdown effect, and siRNA2 had the best knockdown efficiency (*p* < 0.01) (5′-GUGGACUCCAAGCGUUUCUTT-3′), which can be used for subsequent experiments. siRNA-PURB was transfected into C2C12 cells, and siRNA-PURB significantly promoted the expression of *CDK4* and *PCNA* genes at the mRNA level (Figure 6C) (*p* < 0.01). The protein expression level of CDK4 was markedly higher than that of the control group (*p* < 0.01), and the protein expression level of PCNA was significantly increased (Figure 6D,E) (*p* < 0.05). These results indicated that the knockdown PURB promoted the proliferation of C2C12. Similarly, siRNA-PURB was transfected into cells and induced differentiation was performed until myotubes were grown. As shown in Figure 6F, siRNA-PURB significantly upregulated the expression of differentiation biomarker genes (*MyoD*, *MyHC*, and *MyoG*) at the mRNA level (*p* < 0.01). siRNA-PURB markedly elevated the protein expression of MyHC (*p* < 0.01) and significantly promoted the protein expression of MyoD and MyoG (*p* < 0.05) (Figure 6G,H). In conclusion, PURB inhibited the proliferation and differentiation of C2C12 cells.

To explore the effect of PURB on the *TTN* gene, siRNA-PURB was transfected into C2C12 cells, and the effects of PURB on *TTN* during proliferation and differentiation were detected, respectively. The results show that the mRNA and protein levels of siRNA-PURB were significantly higher than those of the control group at the proliferation stage (*p* < 0.01) (Figure 7A–C). siRNA-PURB were markedly increased compared with the control group at the differentiation stage (*p* < 0.01) (Figure 7D), and the protein level was significantly elevated (*p* < 0.05) (Figure 7E,F). These results indicate that PURB inhibited *TTN* gene expression.

### 2.5. circTTN Inhibits TTN Transcription by Recruiting PURB

We intended to verify whether PURB are regulated by circTTN. After circTTN overexpression, PURB induced no significant changes in mRNA and protein levels (Figure 8A–C) (*p* < 0.05). This indicated that circTTN overexpression did not affect the PURB expression level. In a previous study, the effects of circTTN and PURB on the TTN gene were explored, respectively. Then, we utilized the UCSC Genome Browser (http://genome.ucsc.edu/ accessed on 25 August 2022) and PROMO database (https://alggen.lsi.upc.es/cgi-bin/promo_v3/promo/promoinit.cgi?dirDB=TF_8.3 accessed on 25 August 2022) for bioinformatics analysis. We found that the PURB binding motif was present at the *TTN* promoter (Figure 8D), and then, ChIP-qPCR was used to further verify the binding of PURB and the *TTN* promoter. PURB protein significantly bound to the *TTN* promoter (Figure 8E,F) (*p* < 0.01). After circTTN overexpression, we found that PURB protein bound to the *TTN* promoter was significantly higher than in the control group (*p* < 0.01) (Figure 8G). Our study suggested that circTTN inhibited the transcription of the host gene *TTN* by recruiting PURB protein.

## 3. Discussion

Skeletal muscle plays a role in coordinating movement and maintaining normal metabolism in organisms [17]. At present, circRNA is involved in skeletal muscle proliferation and differentiation as a positive or negative regulator of skeletal muscle myogenesis [18]. The C2C12 cell line is a valuable in vitro model, originally monocytes, which can rapidly integrate into multinucleated myotubes after culture with an induced differentiation solution [19]. Due to its unique biochemical characteristics, researchers can better understand the metabolism and differentiation of skeletal muscle at cellular and molecular levels [20]. Sun et al. [21] identified circCSDE with a stable and high expression in porcine skeletal muscle and studied the function of circCSDE on C2C12 cells, where they found that circCSDE1 and its target miR-21-3p jointly regulated the proliferation and differentiation of myoblasts. In their study, the mechanism of circCSDE regulating the growth and development of skeletal muscle in the porcine skeletal muscle injury model was verified. In our study, we first reported that circular intronic RNA circTTN acts as a negative regulator of myogenesis and inhibits the transcription of the host gene *TTN* by recruiting PURB protein in the nucleus (Figure 9).

circRNAs are usually expressed in a tissue-specific manner and perform different biological functions [22]. circRNAs regulate gene expression through the use of a variety of mechanisms. Some studies have shown that circRNA can act as a miRNA sponge or an effective competing endogenouse RNA (ceRNA) molecule [23,24], which is also one of the most classical mechanisms of circRNAs. circRNAs can also bind RBPs. Although the RBPs’ binding site of circRNA is less than that of the corresponding linear mRNA [25], there is still a lot of evidence that RBPs interacts with circRNAs; moreover, they are involved in almost every aspect of the circRNA life cycle, including generation, transcriptional regulation, post-transcriptional regulation, functional execution, specific modifications, and potential extracellular transport pathways [26,27,28]. In this study, we identified the authenticity of circTTN through the use of RNase R digestion and other methods, and then, detected the expression level of circTTN in Ningxiang pig tissues of different days of age and proliferative and differentiated C2C12 cells. It was found that circTTN was highly expressed in the heart and muscle tissues of Ningxiang pig at different periods. It was also differentially expressed in proliferating and differentiated C2C12 cells. This suggested that circTTN may be involved in the regulation of skeletal muscle development. Then, we demonstrated that circTTN inhibits the proliferation and differentiation of C2C12 cells. A previous study has shown that circTTN promotes the proliferation and differentiation of bovine myoblasts [29], suggesting that circTTN may have different effects on the proliferation and differentiation of myoblasts in different mammals. The reason may be that circRNA plays different functions due to different subcellular localization. Our study showed that circTTN is a negative regulator of muscle regeneration in skeletal muscle. Interestingly, we found that circTTN inhibited the expression of the host gene *TTN* during both proliferation and differentiation. In addition, we found that circTTN is mainly located in the nucleus and interacts with PURB protein. Our study showed that PURB appears to be involved in circTTN expression and function.

PURB is a highly conserved protein whose conserved region includes ssDNA and RNA domains. It is a DNA and RNA binding protein involved in the regulation of DNA replication and transcription [30] and can inhibit or promote gene transcription [31]. PURB has been reported to play different roles in many physiological and pathological processes [32]. It can interact with gonadotropin-releasing hormone1 (*GnRH1*) to control fish reproduction by stimulating the pituitary to release gonadotropin-releasing hormone1 [33]. Hariharan et al. [34] demonstrated that PURB protein negatively regulates the trans-activation of smooth muscle *actin* genes in myoblasts. Based on previous studies on PURB, in our study, we confirmed that PURB inhibits the proliferation and differentiation of C2C12 cells and the expression of the *TTN* gene, which was consistent with the function of circTTN.

Previous studies showed that PURB, as a transcription factor, can be recruited by linc-HOXA1 to mediate the transcriptional regulation of lincHOXA1 in embryonic stem cells [35]. A recent study indicated that LncCMRR functionally interacts with the transcriptional suppressor PURB to inhibit its binding potential in the *Flk1* promoter region, thereby protecting *Flk1* transcription during cardiac mesoderm formation [36]. By binding to PURB protein, Circ-Calm4 promoted the transcription of autophagy-related protein *Beclin1*, thereby regulating the hypoxia-induced autophagy of pulmonary artery smooth muscle [37]. circRNAs can not only act as miRNA molecular sponges or translation proteins to regulate muscle growth and development [38] but they also regulate muscle generation by interacting with RBPs. Pandey et al. [14] determined that PURB was distributed in the nucleus and cytoplasm of C2C12 cells, and it combined with circSamd4 located in the nucleus to participate in the transcription regulation of *MHC*, thus promoting myogenesis, which was also the first revelation that circRNA could interact with RBPs to regulate the growth and development of skeletal muscle. Chen et al. [39] pointed out that circMYBPC1 can directly bind MyHC protein, promote the differentiation of myoblasts, and possibly promote skeletal muscle regeneration. One study showed that LncMGPF promoted the expression of HuR in C2C12 cytoplasm, enhancing the stability of *MyoD* and *MyoG* mRNA, and thus promoting myogenesis [40,41]. In this study, we found that PURB and the *TTN* promoter have a binding motif through the use of the UCSC genome browser and PROMO database, and we found that circTTN overexpression promoted the binding of PURB and the *TTN* promoter and inhibited the generation of the *TTN* gene. The protein encoded by the *TTN* gene is the largest known protein and plays a key structural, developmental, and regulatory role in heart and skeletal muscle [42,43]. circTTN inhibits *TTN* transcription through PURB recruitment, which also indicates that circTTN affects skeletal muscle development through PURB recruitment. In addition, our study showed that circTTN or PURB negatively regulated the proliferation and differentiation of C2C12 cells; however, it was unclear whether circTTN inhibited the proliferation and differentiation of myoblasts by recruiting PURB. Furthermore, the role of circTTN in muscle regeneration was not explored in vivo, which is also a limitation of our study. These are something that needs to be addressed in future studies.

## 4. Materials and Methods

### 4.1. Experimental Animals and Samples

All the animals were obtained from Dalong Animal Husbandry Technology Co., LTD., Ningxiang City, Hunan Province. In total, 7 tissue samples (n = 3) were taken from Ningxiang pigs at 30 d, 90 d, 150 d, and 210 d after birth, including heart, liver, spleen, lung, kidney, longissimus dorsi muscle, and adipose. The sampling process strictly followed the asepsis rules. The samples were placed in liquid nitrogen where they were then brought back to the laboratory and finally stored in a freezer at −80 °C. 

### 4.2. Culture and Differentiation of C2C12 Cells

C2C12 cells were purchased from Anweisci (Shanghai, China). The resuscitated cells were cultured with growth medium (CM) in 10 cm dishes. CM was composed of 89% DMEM (Gibco, Waltham, MA, USA), 10% fetal bovine serum (Gibco, Waltham, MA, USA), and 1% penicillin–streptomycin (Gibco, Waltham, MA, USA). When the cell confluence reached 80–90%, trypsin was used for digestion (Gibco, Waltham, MA, USA), and the medium was changed to differentiation medium (DM), which was composed of 2% horse serum and 98% DMEM. C2C12 cells were cultured in a 37 °C incubator containing 5% CO_2_.

### 4.3. RNA Extraction and cDNA Synthesis

An RNA Extraction kit (TIANGEN, Beijing, China) was used to extract total RNA. A nucleic acid concentration was used for the measurements using NanoDrop ND-2000 (Thermo Scientific, Waltham, MA, USA). The samples that needed to be processed with RNase R were digested using RNase R (JISAI, Guangzhou, China). Then, cDNA synthesis was performed using a RevertAid First Strand cDNA Synthesis Kit (Thermo Scientific, Waltham, MA, USA) according to the manufacturer’s instructions.

### 4.4. Quantitative Real-Time PCR

Quantitative real-time PCR (qRT-PCR) primer sequences were designed from the NCBI (https://www.ncbi.nlm.nih.gov/tools/primer-blast/ accessed on 10 November 2021) database, and primers were synthesized from Tsingke Biotechnology (Beijing, China). The primer sequences are shown in Table 2. qRT-PCR was performed using a CFX connect real-time system (Bio-Rad, Hercules, CA, USA), and qRT-PCR data were analyzed using the *β-actin* gene as the reference gene. The relative expression was calculated using the 2^−ΔΔCt^ method.

### 4.5. Vector Construction and siRNA

The circTTN overexpression vector (PCDNA3.1-circTTN) and empty vector (control) were synthesized in JTS Scientific Biotechnology (Wuhan, China). The siRNA and NC of PURB were synthesized in GenePharma Biotechnology (Suzhou, China). Transfection was performed using a Lipofectamine 2000 reagent (Invitrogen, Waltham, MA, USA) according to the manufacturer’s instructions.

### 4.6. CCK8 and EdU Assays

Cell proliferation was detected through Cell Counting Kit-8 (CCK8). The cells were cultured after transfection and inoculated into 96-well plates when the confluence reached 90%. Then, they were cultured for 0 h, 24 h, 48 h, 72 h, and 96 h, respectively, and 10 μL of CCK8 reagent was added 4 h before each assay (Beyotime Biotechnology, Shanghai, China). The absorbance at 450 nm was measured using a microplate reader (Multiskan FC, Thermo Scientific, Waltham, MA, USA). Cell proliferation was detected using the EdU kit (Meilunbio Biotechnology, Shanghai, China). Similarly, the cells were inoculated into 96-well plates and cultured. The EdU solution was diluted with cell growth medium at a ratio of 500:1 to prepare 20 μM of EdU medium. Each well was added with 100 μL of EdU solution, incubated for 2 h, and then fixed with 4% paraformaldehyde, followed by 0.5% tritonX-100 osmotic solution, and finally, stained with EdU and Hoeches 33342.

### 4.7. Flow Cytometry

The cells were transfected when the cells grew to about 60%. After transfection, the cells were mixed with 70% ethanol and fixed at 4 °C for 2 h. Each tube of the sample was stained with 0.5 mL of staining solution (0.5 mL of staining buffer, 10 μL of propidium iodide storage, and 10 μL of RNaseA) and incubated at 37 °C for 30 min in the absence of light. Then, the treated cell suspension was measured using Cytek DxP Athena flow cytometry (Cytek, Fremont, CA, USA).

### 4.8. Immunofluorescence Staining

The cells were fixed with 4% paraformaldehyde for 30 min, incubated with 0.5% triton X-100 at room temperature for 20 min, and closed with 5% BSA (Bio Froxx, Frankfurt, Germany) for 2 h. Anti-MyHC monoclonal antibody was used for diluting 5% BAS (1:200, DSHB, Iowa, IA, USA) and was incubated overnight at 4 °C. Then, it was incubated with anti-mouse IgG (1:1000, Abbkine, Wuhan, China) for 2 h. Next, nuclei were stained using DAPI (Solarbio, Beijing, China) for 10 min without light. Finally, photographs were taken and analyzed under an inverted microscope (Axio Vert A1, ZEISS, Oberkochen, Germany).

### 4.9. Western Blot Analysis

The cell proteins were extracted with RIPA Lysis Buffer (Beyotime Biotechnology, Shanghai, China), and the protein concentration was quantified using a BCA protein quantitative/concentration assay kit (Meilunbio Biotechnology, Shanghai, China). The denatured protein samples were electrophoreted in Tris/Glycine/SDS Buffer (Epizyme, Biotechnology, Shanghai, China), and then, the proteins were transferred to the PDVF membrane for electrotransfer in the electrotransfer solution (Yamei, Biotechnology, Shanghai, China). The membranes were sealed with the sealing liquid for 2 h, and then β-actin (1:1000, TransGen, Beijing, China), CDK4 (1:1000, AiFang, Changsha, China), PCNA (1:2000, ZENGBIO, Chengdu, China), CCND (1:1000, CST, Boston, MA, USA), MyoD (1:1000, HUABIO, Hangzhou, China), MyoG (1:200, DSHB, Iowa, IA, USA), MyHC (1:1000, DSHB, Iowa, IA, USA), Titin (1:1000, Bioword, Minnesota, MN, USA), and PURB (1:1000, Proteintech, San Diego, CA, USA) antibodies were incubated at 4 °C overnight. Finally, the membranes were incubated with the secondary antibody (1:10,000, ZENBIO, Chengdu, China) for 2 h, and the protein bands were developed using an ECL chemiluminescent substrate kit (Biosharp, Guangzhou, China). All the protein levels were analyzed by β-actin protein reference normalization.

### 4.10. Nucleoplasmic Separation and RNA Fluorescence In Situ Hybridization

Nuclear and cytoplasmic RNA was extracted using a cytoplasmic and nuclearRNA purification kit (Norgen Biotek, Thorold, ON, Canada) and was quantified using qRT-PCR analysis. We made cell creep tablets, fixed them with 4% paraformaldehyde, and sent them to Pinuofei Biotechnology (Wuhan, China) for a RNA fluorescence in situ hybridization (FISH) experiment. The probe sequence of circTTN is 5′-UGAGGCUGCCAACCGGAACAAUGACG-3′.

### 4.11. RNA Pull-Down and Mass Spectrometry

Biotin-labeled circTTN probe and NC probe were provided by Ribo Biotechnology (Guangzhou, China). Pull-down protein experiments were performed using an RNA pulldown kit (BersinBio, Guangzhou, China). First, the circTTN probe and NC probe formed a secondary structure and were incubated with streptavidin magnetic beads, respectively. Then, 2 × 10^7^ cell samples were collected for protein extraction and the nucleic acid of protein samples were removed, and then, the probe–magnetic bead complex was rotatively incubated with cell extracts. Finally, the histone samples of the circTTN group and NC group were collected, and silver dyeing was performed with a silver dyeing kit (Beyotime, Shanghai, China). First, 30% ethanol was used for fixing, and then, silver sensitizing solution, silver solution, silver dyeing solution, and silver dyeing stop solution were added successively. The silver dyed adhesive strips were photographed and the results were analyzed. RNA pulldown protein samples were determined and analyzed using mass spectrometry in Genecreate Biotechnology (Wuhan, China).

### 4.12. RNA Immunoprecipitation (RIP) Analysis

An RNA Immunoprecipitation Kit (BersinBio, Guangzhou, China) was used for RIP analysis. First, 2 × 10^7^ cells were lysed to remove the DNA, and then the cell lysate samples were divided into 3 parts according to 0.8 mL (IP), 0.8 mL (IgG), and 0.1 mL (Input). IP and IgG samples were added with the corresponding experimental antibodies PURB (ProteintechSan Diego, CA, USA) and IgG (BersinBio, Guangzhou, China), respectively, and were incubated at 4 °C for 16 h. Then, they were rotatively incubated with balanced protein A/G beads for 1 h, followed by RNA eluting using an elution buffer. Finally, RNA was extracted with a mixture of phenol–chloroform–isoamyl alcohol, which was detected and analyzed using qRT-PCR according to the percentage input method.

### 4.13. ChIP-qPCR Analysis

A SimpleChIP^®^ Plus Enzymatic Chromatin IP Kit (CST, Boston, MA, USA) was used for ChIP analysis. In brief, the cells were crosslinked with 1% formaldehyde for 10 min and incubated with 10× glycine for 5 min, followed by the addition of protease enzyme (PIC); then, chromatin fragment and diluted chromatin were added to the corresponding IP (PURB) and IgG immunoprecipitation antibody. They were then incubated overnight by rotation at 4 °C, followed by the addition of protein G magnetic beads. After incubation at 4 °C for 2 h, the DNA complex was cleaned with high-salt and low-salt washing solution, and chromatin was eluted with a ChIP Elution buffer and Proteinase K was added. Finally, DNA purification was performed. The combination of PURB and the *TTN* promoter were analyzed using qRT-PCR according to the percentage input method.

### 4.14. Statistical Analysis

We used Microsoft Excel to sort the data. We analyzed all the data using *t*-tests and a one-way ANOVA and calculated qRT-PCR data using the 2^−ΔΔCt^ method. In addition, we analyzed RIP-qPCR and ChIP-qPCR using the percentage input method, counted the number of cells using Image J software (version 1.49), and mapped these results using GraphPad Prism 8.0 software. Data analysis was expressed as mean ± SEM, n = 3. * *p* < 0.05 and ** *p* < 0.01 denote a significant and extremely significant mean, respectively.

## 5. Conclusions

In conclusion, we verified a novel circular intronic RNA circTTN and validated its function in C2C12 cells. Our study shows that circTTN inhibits myoblast proliferation and differentiation and influences skeletal muscle development through the regulation of transcription using the circTTN/PURB/TTN axis. This study provides new insights and references for future comprehensive analyses of skeletal muscle growth and development.

## Figures and Tables

**Figure 1 ijms-24-09859-f001:**
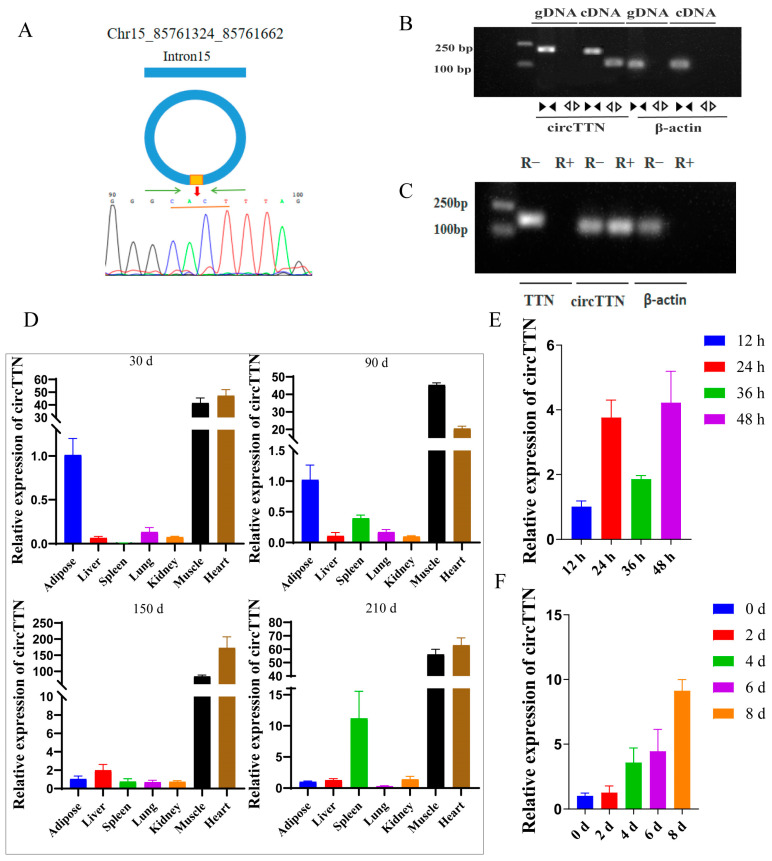
Characterization analysis of circTTN. (**A**) circTTN source diagram and the joint sequence by sanger sequencing. (**B**) The authenticity of circTTN was detected by divergent primer and convergent primer. (**C**) The stability of circTTN was determined by RNase R digestion. (**D**) The expression of circTTN in different tissues of Ningxiang pig at 30 d, 90 d, 150 d, and 210 d after birth was detected. (**E**) circTTN expression in C2C12 cells during proliferation period was detected. (**F**) circTTN expression during C2C12 cells differentiation was detected.

**Figure 2 ijms-24-09859-f002:**
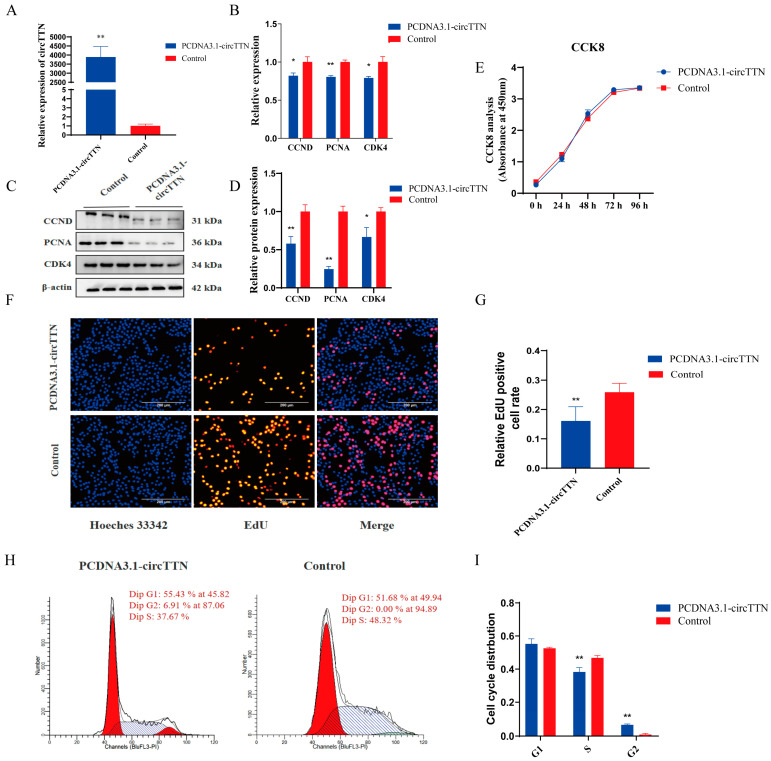
Effect of overexpression of circTTN on proliferation of C2C12 cells. (**A**) Transfection efficiency was screened by qRT-PCR 24 h after transfection with circTTN overexpression (PCDNA3.1-circTTN) and empty vector (control). The transfection condition is 4.0 μg PCDNA3.1-circTTN/control and 10 μL Lipofectamine 2000 reagent. (**B**) The effect of circTTN overexpression was detected by qRT-PCR on proliferation biomarker genes at the mRNA level. (**C**) The effect of circTTN overexpression was detected by Western blot on proliferation biomarker genes at the protein level. (**D**) Gray scale analysis of protein band in B. (**E**) Detection of cell proliferative activity using CCK8 after transfection with PCDNA3.1-circTTN and control. (**F**) Cell proliferation was detected by EdU. Scale bars indicate 200 μm. (**G**) EdU-positive cells were counted. (**H**) Cell cycle changes were detected using flow cytometry. (**I**) Count the number of proliferating cells during the cell cycle. * *p* < 0.05 and ** *p* < 0.01. Results are presented as mean ± S.E.M. n = 3.

**Figure 3 ijms-24-09859-f003:**
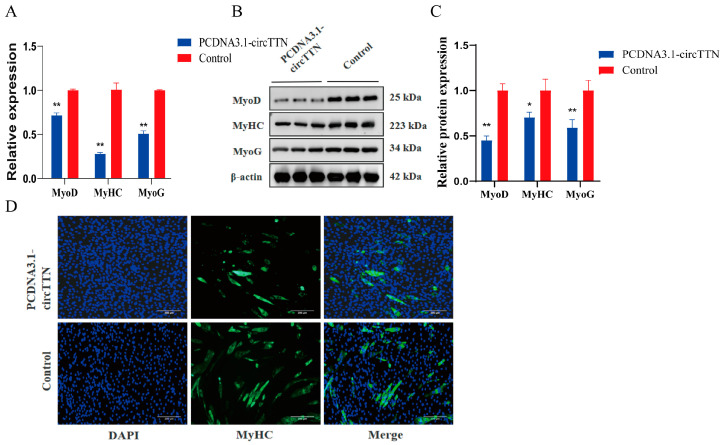
Effect of overexpression of circTTN on differentiation of C2C12 cells. (**A**) At 72 h after transfection and differentiation, qRT-PCR was used to detect the effect of circTTN overexpression on the mRNA level of differentiation biomarker genes. (**B**) Effect of overexpression of circTTN on differentiation biomarker genes at the protein level. (**C**) Gray scale analysis of protein band in B. (**D**) The changes in circTTN overexpressed myotubes were detected by immunofluorescence at sixth day of differentiation. Scale bars indicate 200 μm. * *p* < 0.05 and ** *p* < 0.01. Results are presented as mean ± S.E.M. n = 3.

**Figure 4 ijms-24-09859-f004:**
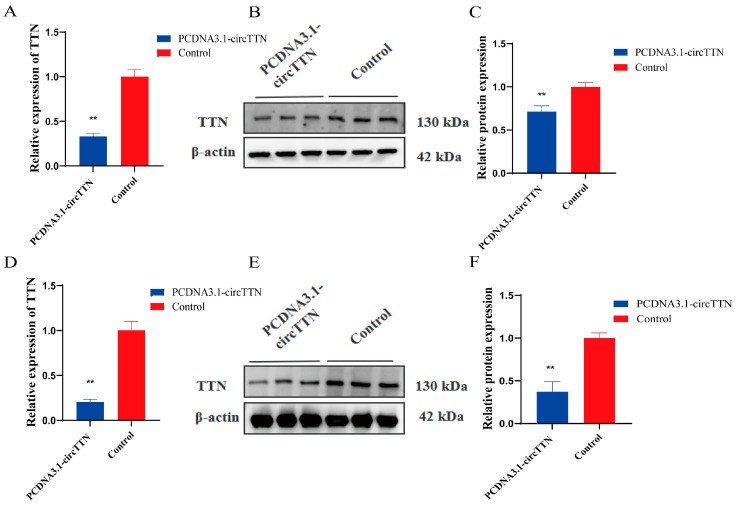
Effect of overexpression of circTTN on *TTN* gene. (**A**) qRT-PCR assay was used to detect the effect of overexpressed circTTN on the proliferation stage of *TTN*. (**B**) Western blot assay was used to detect the effect of overexpressed circTTN on *TTN*. (**C**) Gray scale analysis of protein bands in figure B. (**D**) The effect of circTTN overexpression on *TTN* differentiation stage was detected by qRT-PCR. (**E**) Western blot assay was used to detect the effect of overexpressed circTTN on *TTN* at differentiation stage. (**F**) Protein band gray analysis statistics in figure (**E**). ** *p* < 0.01. Results are presented as mean ± S.E.M. n = 3.

**Figure 5 ijms-24-09859-f005:**
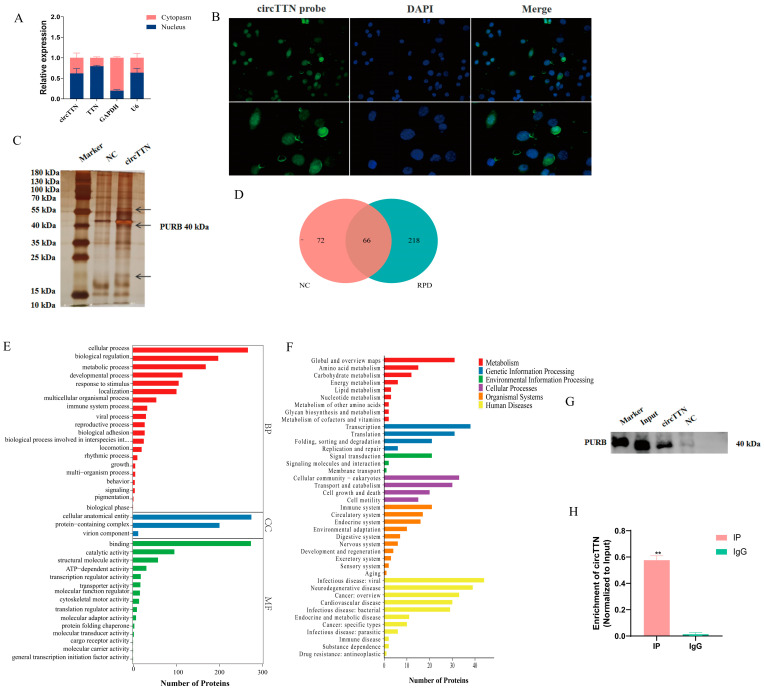
circTTN interacts with PURB protein. (**A**) The location of circTTN was determined by nucleoplasmic separation. (**B**) We determined the location of circTTN using FISH. 400× (**upper** images) and 1000× (**bottom** images) (**C**) In the RNA pull-down experimental silver glue diagram, arrows indicate differential proteins. (**D**) Protein statistics were identified by mass spectrometry. (**E**) Functional enrichment of protein GO was identified by mass spectrometry. (**F**) Protein KEGG analysis was identified by mass spectrometry. (**G**) The expression of Pur-beta (PURB) protein in RNA pull-down eluent was detected using WB. (**H**) RIP experiments reverse demonstrated that PURB protein interacts with circTTN. ** *p* < 0.01. Results are presented as mean ± S.E.M. n = 3.

**Figure 6 ijms-24-09859-f006:**
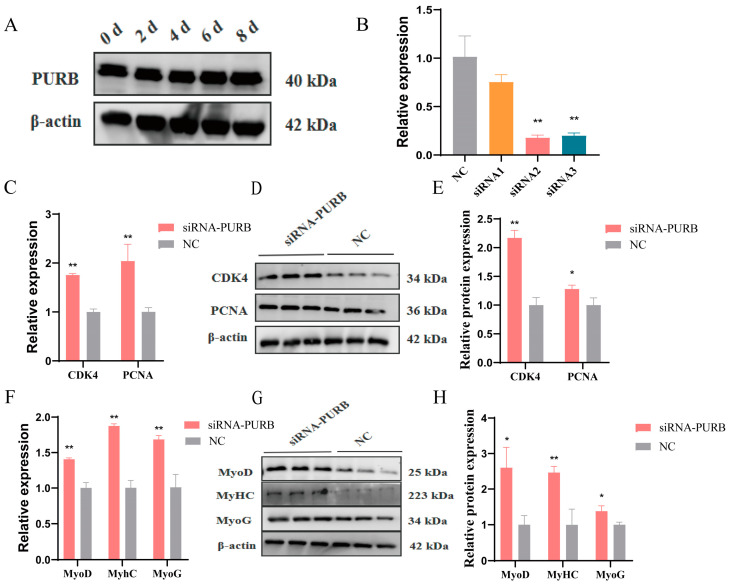
Knockdown effect of PURB on proliferation and differentiation of C2C12 cells. (**A**) PURB protein expression in naturally differentiated C2C12 cells. (**B**) Screening of siRNA sequences by qRT-PCR 24h after transfection with siRNA-PURB and NC. The transfection conditions are 100 pmol siRNA-PURB/NC and 5 μL Lipofectamine 2000 reagent. (**C**) The effect of siRNA-PURB proliferation biomarker genes at mRNA level. (**D**) The effect of siRNA-PURB on proliferation biomarker genes at the protein level. (**E**) Gray scale analysis of protein bands in (**D**). (**F**) At 72 h after transfection and differentiation, qRT-PCR was used to detect the effect of siRNA-PURB on the mRNA level of differentiation biomarker genes (**G**) The effect of siRNA-PURB on differentiation biomarker genes at the protein level. (**H**) Gray scale analysis of protein bands in Figure (**G**). * *p* < 0.05 and ** *p* < 0.01. Results are presented as mean ± S.E.M. n = 3.

**Figure 7 ijms-24-09859-f007:**
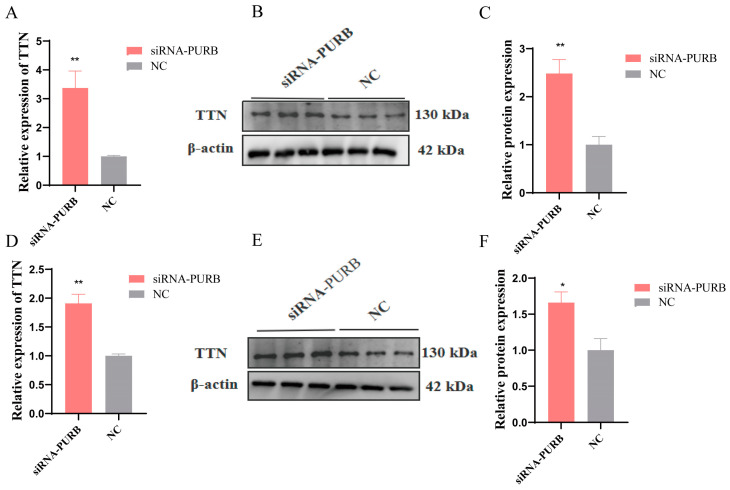
Effect of PURB knockdown on *TTN* gene. (**A**) qRT-PCR was used to detect the effect of PURB knockdown on *TTN* during proliferation stage. (**B**) Western blot assay was used to detect the effect of PURB knockdown on *TTN* during proliferation stage. (**C**) Gray scale analysis of protein bands in (**B**). (**D**) qRT-PCR assay was used to detect the effect of PURB knockdown on *TTN* during differentiation. (**E**) Western blot assay was used to detect the effect of PURB knockdown on *TTN* during differentiation. (**F**) Protein band gray analysis statistics (**E**). * *p* < 0.05 and ** *p* < 0.01. Results are presented as mean ± S.E.M. n = 3.

**Figure 8 ijms-24-09859-f008:**
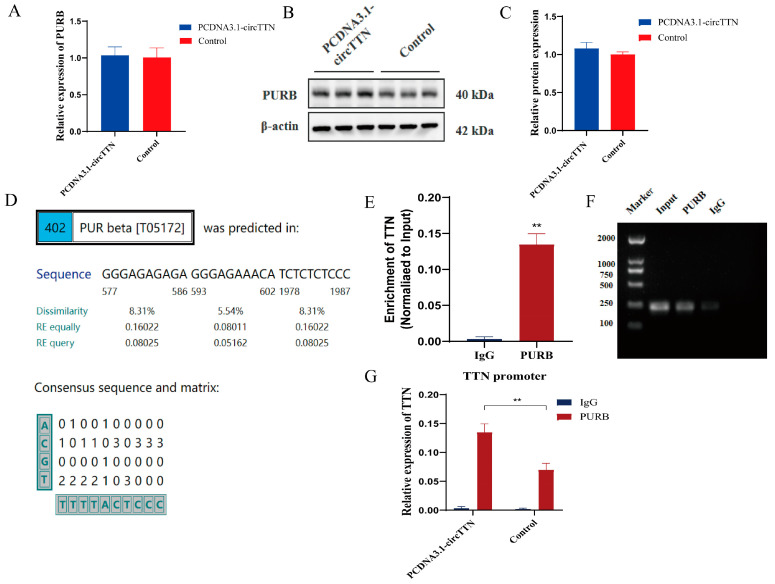
circTTN inhibits *TTN* gene expression by recruiting PURB. (**A**) The effect of overexpression of circTTN on PURB mRNA level. (**B**) The effect of overexpression of circTTN on PURB at the protein level. (**C**) Gray scale analysis of protein bands in (**B**). (**D**) PROMO database predicted PURB binding to *TTN* promoter. (**E**) ChIP-qPCR showed that PURB antibody was enriched to *TTN* promoter. (**F**) Glue map of ChIP-qPCR demonstrated that PURB antibody was enriched to *TTN* promoter. (**G**) ChIP-qPCR showed that circTTN overexpression promoted PURB and *TTN* promoter aggregation. ** *p* < 0.01. Results are presented as mean ± S.E.M. n = 3.

**Figure 9 ijms-24-09859-f009:**
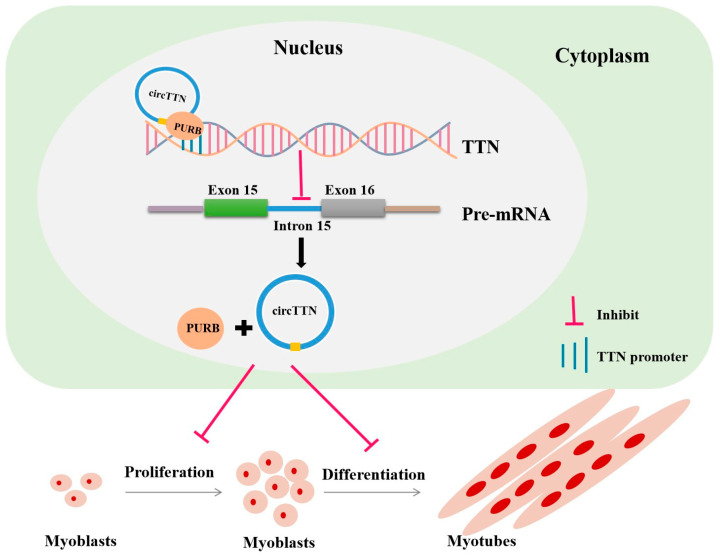
Molecular regulatory mechanism model of circTTN. Circular intronic RNA circTTN acts as a negative regulator of myogenesis and inhibits transcription of host gene TTN by recruiting PURB protein in the nucleus.

**Table 1 ijms-24-09859-t001:** Possible circTTN interaction proteins identified by RNA pull-down RPD group (top 30).

Accessions	Names	Unused	Description
Q6QAQ1	ACTB	191.97	Actin, cytoplasmic 1
P02543	VIM	149.62	Vimentin
A0A287AU59	ACACA	133.63	Acetyl-CoA carboxylase 1
A0A287BNK7	PLEC	133	Uncharacterized protein
A0A287BHM1	TPM1	91.59	Tropomyosin alpha-1 chain
A0A5G2RI50	MYO1C	83.87	Unconventional myosin-Ic isoform b
Q2XVP4	TUBA1B	61.55	Tubulin alpha-1B chain
P02554	Tubulin beta	60.54	Tubulin beta chain
F6Q364	ACTA2	59.23	Uncharacterized protein
P02540	DES	52.77	Desmin
A0A287BQP1	RBM25	36.79	Uncharacterized protein
A0A288CG57	EEF1A1	36.1	Elongation factor 1-alpha
A0A286ZVS0	SF3B3	35.56	Splicing factor 3B subunit 3
F1SSA6	MYH10	33.34	Myosin motor domain-containing protein
A0A287BSP1	MYO5A	32.65	Uncharacterized protein
A0A287A4H2	PURB	32.47	Transcriptional activator protein Pur-beta
P80021	ATPATP5F1A	31.56	ATP synthase subunit alpha, mitochondrial
F1RUL1	RBM17	29.87	Splicing factor 45
A0A287B1F6	MPRIP	29.21	Uncharacterized protein
K9J4V0	SNRNP200	29	Protogenin isoform X1
I3L920	CORO1C	27.2	Coronin
I3L650	CALD1	26.54	Uncharacterized protein
A0A287BIL8	GN = HSPA5	26.3	78 kDa glucose-regulated protein
K7GLT8	ATP5F1B	24.2	ATP synthase subunit beta
A0A5G2R8T3	MYO1E	23.9	Uncharacterized protein
A0A286ZHW0	SF3B1	21.89	Splicing factor 3B subunit 1 isoform 1
F1S5A8	DHX15	21.49	RNA helicase
A0A286ZWK2	HSPA8	20.98	Uncharacterized protein
I3LLT2	CAD	20.86	Aspartate carbamoyltransferase
A0A286ZXU2	FLNA	19.38	Uncharacterized protein

**Table 2 ijms-24-09859-t002:** Primer sequences information.

Genes	Primer (5′→3′)	Length
circTTN (convergent primer)	F: TGTCTCACTGCCTTGTCTGATG	241 bp
	R: TTAGCAGCTGGTTCAGTCACC	
circTTN (divergent primer)	F: AGCTGCTAAAGTGCCCATTCC	139 bp
	R: AAACAAAGAACATCAGACAAGGCA	
*β-actin*	F: GGACTGTTACTGAGCTGCGTT	290 bp
	R: CGCCTTCACCGTTCCAGTT	
*CDK4*	F: CGAGCGTAAGGCTGATGGAT	177 bp
	R: CCAGGCCGCTTAGAAACTGA	
*PCNA*	F: GCCGAGACCTTAGCCACATT	229 bp
	R: GTAGGAGACAGTGGAGTGGC	
*CCND*	F: TCAAGTGTGACCCGGACTG	235 bp
	R: GCTCCTTCCTCTTTGCGGG	
*MyoD*	F: AGTGAATGAGGCCTTCGAGA	169 bp
	R: GCATCTGAGTCGCCACTGTA	
*MyoG*	F: CAATGCACTGGAGTTCGGT	134 bp
	R: CTGGGAAGGCAACAGACAT	
*MyHC*	F: CGGTCGAAGTTGCATCCCT	141 bp
	R: GAGCCTCGATTCGCTCCTTT	
*TTN*	F: GTCCTCCATCTCCTCCTGGT	173 bp
	R: GCGTTTTGTTGACCCTCACC	
*PURB*	F: GGAGCGACAGAGGGATAAGC	181 bp
	R: TAGCTCACTGGGGACGAGAA	
*TTN*-promoter	F: TGCCCACAAGTGTCTTCAACT	179 bp
	R: ACCTGTTCCTCTGATTGGCAG	

## Data Availability

The data analyzed during the current study are available from the corresponding author on reasonable request.

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
