# Peer review of "Circular Intronic RNA circTTN Inhibits Host Gene Transcription and Myogenesis by Recruiting PURB Proteins to form Heterotypic Complexes"

_ijms, 2023, doi:10.3390/ijms24129859_

Round 1
Reviewer 1 Report
In this manuscript, Ai and colleagues describe a role for the noncoding, circular RNA circTTN as a novel regulator of myogenesis trough its interaction with the transcriptional regulator PURB. Based on previous studies demonstrating that circTTN inhibits myoblast proliferation and differentiation, the authors performed RNA pull-down and mass spectrometry experiments and identified PURB as one of the main interacting partners of circTTN. Transient, plasmid-based overexpression of circTTN in C2C12 cells resulted in the reduced proliferation (measured by CCK8 and EdU assays, by flow cytometry and by RT-qPCR and Western blotting on CCND, CDK4 and PCNA biomarkers, Figure 2) and differentiation of cells, indicated by the reduced expression of MyoD, MyHC, and MyoG biomarkers (Figure 3). Overexpression of circTTN also affected the expression of TTN gene, as a reduced expression was observed in RT-qPCR experiments (Figure 4). RNA pulldown-experiments followed by proteome analysis identified 218 proteins that are enriched in the circTTN samples vs control (figure 5 and table 1). Among these, the authors decided to focus on PURB. The interaction between circTTN and PURB was confirmed in vitro, and the reduced expression of PURB by siRNA in C2C12 cells resulted in the increased proliferation and differentiation of those cells, demonstrated by a similar set of experiments mentioned before (Figure 6). Finally, by ChIP-qPCR experiments, the authors demonstrate that PURB bind the TTN promoter (Figure 8). All these experiments led the authors to hypothesize a molecular regulatory mechanism model in which circTTN inhibits TTN transcription by recruiting PURB.
In principle, this manuscript is interesting for the broad readership of IJMS.
However, I think the authors’ conclusions are not fully supported by their results, as some crucial experiment or control are missing; this makes the manuscript difficult to accept in the current form.
To summarize: 1) overexpression of circTTN inhibits proliferation and differentiation of C2C12 cells; 2) knockdown of PURB promoted the proliferation of C2C12; 3) circTTN interacts with PURB protein; 4) the proposed model is that circTTN acts as a negative regulator of myogenesis and inhibits transcription of host gene TTN by recruiting PURB protein on the TTN promoter. This is reasonable, and the most relevant data supporting this hypothesis are the ChIP-qPCR experiments shown in Figure 8. However, I do not fully understand the panel E of Fig. 8 and the authors’ statement: “ChIP-qPCR showed that PURB antibody was enriched to TTN promoter”. If I correctly remember the principles of ChIP-qPCR, after protein crosslinking to DNA and chromatin fragmentation, an antibody specific for a target protein (here PURB) is used for immunoprecipitation. The pulled-down DNA is then sequenced by NGS, (if we want to identify all target sequences of PURB) or used as a template in PCR experiment by using primers specific for a target DNA, if we want to confirm a specific interaction (whose results are shown in panels F and G). Therefore, what is enriched is not the antibody, but the TTN promoter DNA.
In Figure 5, panel A, the authors present the results from a nucleus/cytoplasm fractionation experiment showing that in C2C12 cells overexpressing circTTN, the latter is (not surprisingly) mostly localized in the nucleus. Indeed, the prevalent nuclear localization of TTN mRNA is surprising, since it is a coding RNA and therefore should me mostly cytoplasmatic. Is this a consequence of circTTN overexpression? If so, it could explain itself (without the involvement of PURB) the reduced expression of TTN protein and, consequently, the reduced proliferation and differentiation of C2C12 cells. The authors should address this alternative hypothesis by comparing the nucleus/cytoplasm localization of TTN mRNA in cells transfected with circTTN versus control cells.
Another concern is why to choose to validate the interaction between PURB and circTTN, among the 218 proteins identified in the screening. Was it the most enriched? Or, more simply, best fitted with the proposed mechanism? The authors should explain the rationale for choosing only PURB for subsequent validations of RNA pulldown experiments.
The authors always mentioned TTN using the gene symbol. I suppose that the expression product of TTN is the protein titin. Titin is a giant protein (around 3 MDa), how can be possible that the authors see it as a 130kDa band in their Western blots? Could they add some comments?
Another crucial point is that several information on the circular RNA plasmid used in this study are missing. How does the pCDNA3.1-circTTN vector work? Did the authors check for the efficiency of circularization of expressed RNA? Figures 2A show that in transfected cells the level of circTTN is thousands of times higher than untransfected ones. How much RNA is indeed linear? Did the authors check?
Minor comments:
Authors should put more efforts into proofreading their manuscript, it would help the reviewer's work by saving time, so I would be able to prepare my review timely. To give more examples:
Line 53: “In this study, we previously found a circular intronic RNA circTTN produced by the TTN gene through high-throughput sequencing”. This is contradictory: “previously” means “previously published research”. “In this study” should be removed.
Line 93: the statement “proliferating genes” is not correct, it should be “genes promoting cell proliferation” or “proliferation biomarker genes”.
Line 156: “The results showed that circTTN and TTN genes existed in the nucleus and cytoplasm”. Genes do not exist in the cytoplasm (unless we consider mitochondria) but only in the nucleus; the products of their transcription (RNAs) can be localized either in the nucleus or cytoplasm.
Line 193: the paragraph is entitled “2.4. PURB inhibited proliferation and differentiation of C2C12 cells and TTN gene”. This is an indirect conclusion from the interpretation of PURB siRNA experiments. A title like “PURB knockdown promotes proliferation and differentiation of C2C12 cells and increase TTN expression” would be more appropriate.
The authors did not mention one single time that TTN gene is titin protein. They should name titin when refer to experiments involving proteins (e.g., western blotting). Moreover, in all Western blots (Figs. 3B, 6D, 6G, 8B), for each condition (experiment vs control) three lanes are shown. I supposed they represent three replicas. Each lane should be labelled.
Table 1: what “unused” stans for?
“Extremely significantly higher” is redundant. “higher” or “increased” would be sufficient: statistical significance is already provided in the figures.
Line 396: “Cell proliferation was detected through CCK8”. CCK8 is the name of a kit, not a technique. Cell proliferation was detected by using a commercial Cell Counting Kit – 8 (CCK8)
The manuscript should be proofread to amend some grammatical mistakes or misconceptions
Few examples: "proliferating genes" should be “genes promoting cell proliferation”
“The results showed that circTTN and TTN genes existed in the nucleus and cytoplasm”. These are not genes, but transcripts.
Reviewer 2 Report
In the manuscript entitled “Circular intronic RNA circTTN inhibits host gene transcription and myogenesis by recruiting PURB proteins to form hetero typic complexes”, the authors investigated circTTN on myoblast growth and the potential molecular mechanism. Their results indicate that circTTN inhibits myoblast proliferation and differentiation. Mechanistically, circTTN blocks its host gene TTN expression by recruiting PURB proteins to form heterotypic complexes. Although the authors provided a fair amount of data, the study has limited novelty due to a similar work published in 2019. In addition, most of the experiments were performed using a single cell line C2C12, making the conclusion less convincing.
1. In Figure 2A, the data indicated the circTTN was overexpressed approximately 4000 times compared to its control. The level is much more than physiological conditions. In this scenario, circTTN overexpression may lead to off-target effects and artificial conclusions, such as cell cycle arrest.
2. The figure legends are too concise. It would be better to include the experimental details, such as time points, doses, etc.
3. The conclusion that circTTN inhibits myoblast growth is solely from a mouse cell line. At least the results need to be validated using one or two more cell lines/primary cells.
4. In addition to in vitro experiments, it would be recommended to perform additional in vivo experiments. For instance, does circTTN change during muscle regeneration?
5. A controversial conclusion was reported in a similar study. (Mol Ther Nucleic Acids. 2019 Dec 6;18:966-980. doi: 10.1016/j.omtn.2019.10.019. Epub 2019 Oct 25.) However, the authors stated that circTTN may have different effects on the proliferation and differentiation of myoblasts in different mammals. If so, the study of circTTN using mouse C2C12 has no relevance to humans.
Reviewer 3 Report
This is a nice work on studying the role of circTTN in myogenesis and its mechanism. The authors provide sufficient data to support their conclusion and the results are well presented. However, the writing of this manuscript needs careful proofreading and some minor problems need to be addressed as follows.
1. In Figure 1D and 1E, the authors analyzed the expression of circTTN in pigs and cells along with time. What is the implication of these results?
2. In Figure 2B and 2D, there is a discrepancy between the mRNA expression and protein level of CCND and PCNA. Why is the relative expression of mRNA around 80% but the relative protein level below 50%?
3. In Figure 2E, what is the unit of the Y-axis?
4. In Figure 3, there is still a discrepancy between the relative mRNA expression and protein level of MyoD, MyHC, and MyoG. The authors need to provide possible reasons.
5. In all the confocal images, a scale bar needs to be provided within the image.
6. It would be a good idea to put Figure 8 at the beginning of the manuscript for readers to better understand this work.
The overall writing is good and clear but still needs careful proofreading. For example, "extremely" and "significantly" are not usually put together to describe the results.
Round 2
Reviewer 1 Report
No more comments
Author Response
Thank you very much for taking your time to review our manuscript "Circular intronic RNA circTTN inhibits host gene transcription and myogenesis by recruiting PURB proteins to form heterotypic complexes" (ijms-2397676). We really appreciate all your professional comments and suggestions! We have studied comments carefully and have made correction which we hope meet with approval (see below). Revised portions are marked in red in the paper. We believe that the manuscript has been considerably improved after the revisions and hope it is suitable for publication in International Journal of Molecular Sciences.
Response to Academic Editor
Point 1. In addition to in vitro experiments, reviewer 2 recommended performing additional in vivo experiments to prove that circTTN changes during muscle regeneration.
Response 1: Thank you very much for your professional comments. We strongly believe that in vivo experiments to explore the role of circTTN in the process of muscle regeneration would make the paper more complete and convincing, which is also a limitation of our study. We added a discussion of this flaw to the manuscript, please check line 372-373.
Point 2. A controversial conclusion was reported in a similar study. (Mol Ther Nucleic
Acids. 2019 Dec 6;18:966-980. doi: 10.1016/j.omtn.2019.10.019. Epub 2019 Oct 25.) and the comments of the author's are convincing. According to the authors, circTTN may have different effects on the proliferation and differentiation of myoblasts in different mammals. This reinforces the idea that the study of circTTN using mouse C2C12 has no relevance to humans.
Response 2: Thank you very much for your professional comments. We stated the idea that circRNAs differ in subcellular localization and so may have different roles in proliferation and differentiation in different mammals. We have added to this argument in the manuscript. Please check line 319-320. We used circBASE (http://www.circbase.org/) for conservation analysis of circTTN sequences. We found that circTTN sequence is highly conserved with the human sequence (hsa_circ_0057222 Position chr2: 179544325-179549476), so we hypothesize that it has a similar function in human myoblasts.
Thank you again and look forward to hearing from you soon.
Yours Sincerely,
Name: Haiming Ma, Yulong Yin
E-mail: mahaiming2000@163.com; yinyulong@isa.ac.cn
College of Animal Science and Technology, Hunan Agricultural University
Guangdong Laboratory for Lingnan Modern Agriculture, Guangzhou 510642, China
Institute of Subtropical Agriculture, Chinese Academy of Sciences, Changsha 410125, China.

Reviewer 2 Report
Unfortunately, I cannot see substantial improvement in the revised manuscript given that the authors did not address my concerns regarding reproducibility and scientific rigors.
Author Response

(The authors gave the same response as above.)
